# Bioinformatical Design and Performance Evaluation of a Nucleocapsid- and an RBD-Based Particle Enhanced Turbidimetric Immunoassay (PETIA) to Quantify the Wild Type and Variants of Concern-Derived Immunoreactivity of SARS-CoV-2

**DOI:** 10.3390/biomedicines11010160

**Published:** 2023-01-08

**Authors:** Leoni Wey, Thomas Masetto, Alexander Spaeth, Jessica Brehm, Christian Kochem, Marco Reinhart, Holger Müller, Uwe Kempin, Franziska Lorenz, Christoph Peter, Matthias Grimmler

**Affiliations:** 1DiaSys Diagnostic Systems GmbH, Alte Str. 9, 65558 Holzheim, Germany; 2Hochschule Fresenius Gem. Trägergesellschaft mbH, University of Applied Sciences, Limburger Str. 2, 65510 Idstein, Germany; 3Institut für Molekulare Medizin I, Heinrich-Heine-Universität, Universitätsstr. 1, 40225 Düsseldorf, Germany; 4MVZ Medizinische Labore Dessau Kassel GmbH, Bauhüttenstr. 6, 06847 Dessau-Roßlau, Germany; 5GfA GmbH, Allgäuer Str. 1, 87459 Pfronten, Germany; 6pes Medizinische Diagnosesysteme GmbH, Hauptstr. 103, 04416 Markkleeberg, Germany

**Keywords:** SARS-CoV-2, COVID-19, VOC, S-RBD, N protein, PETIA, performance evaluation

## Abstract

Since SARS-CoV-2 emerged in December 2019 in Wuhan, the resulting pandemic has paralyzed the economic and cultural life of the world. Variants of concern (VOC) strongly increase pressure on public health systems. Rapid, easy-to-use, and cost-effective assays are essential to manage the pandemic. Here we present a bioinformatical approach for the fast and efficient design of two innovative serological Particle Enhanced Turbidimetric Immunoassays (PETIA) to quantify the SARS-CoV-2 immunoresponse. To confirm bioinformatical assumptions, an S-RBD- and a Nucleocapsid-based PETIA were produced. Sensitivity and specificity were compared for 95 patient samples using a BioMajesty™ fully automated analyzer. The S-RBD-based PETIA showed necessary specificity (98%) over the N protein-based PETIA (21%). Further, the reactivity and cross-reactivity of the RBD-based PETIA towards variant-derived antibodies of SARS-CoV-2 were assessed by a quenching inhibition test. The inhibition kinetics of the S-RBD variants *Alpha*, *Beta*, *Delta*, *Gamma*, *Kappa*, and *Omicron* were evaluated. In summary, we showed that specific and robust PETIA immunoassays can be rapidly designed and developed. The quantification of the SARS-CoV-2-related immunoresponse of variants (*Alpha* to *Kappa*) is possible using specific RBD assays. In contrast, *Omicron* revealed lower cross-reactivity (approx. 50%). To ensure the quantification of the *Omicron* variant, modified immunoassays appear to be necessary.

## 1. Introduction

Severe acute respiratory syndrome coronavirus 2 (SARS-CoV-2) emerged in December 2019 [1,2], raising a major challenge for health systems and the economy worldwide. SARS-CoV-2 is the etiologic agent of the pandemic coronavirus disease 2019 (COVID-19) [3], a severe acute respiratory syndrome including pneumonia with fever and a strong cough, as well as a hyperinflammatory response, vascular damage, and widespread thrombosis [4].

SARS-CoV-2 is a positive-sense single-stranded RNA virus and a member of the family *Coronaviridae* [5,6]. The virus exhibits a high similarity with the two human infecting *Beta* coronaviruses, SARS-CoV-1 and MERS-CoV, and partial similarities with the *Beta* coronaviruses HKU1-CoV and OC43-CoV and the *Alpha* coronaviruses 229E-CoV and NL63-CoV [6,7]. The last four species circulate endemically worldwide, predominantly causing mild colds but sometimes also severe pneumonia, especially in early childhood and in the elderly. Consequently, the specific detection and quantification of SARS-CoV-2, unaffected by acute or previous infection with other coronaviruses, is a prerequisite for specific and reliable serological SARS-CoV-2 test systems [8]. Coronaviruses (CoVs) encode four important structural proteins that are required to produce a structurally complete virus particle: the Spike protein (S), the Nucleocapsid protein (N), the Membrane protein (M), and the Envelope protein (E) (Figure 1a). The details of the structural similarity of the main structural proteins S, S-RBD, and N of SARS-CoV-2 in comparison to the homologous proteins of the other coronaviruses are shown in Table 1. The S protein on the surface of the virus is the most important structural component since it is involved in infection [9,10]. It is a large membrane-anchored protein that assembles to form trimers on the surface of the virus via its S2 subunit (crown-like appearance). Each Spike monomer contains a receptor-binding domain (RBD) in the N-terminal S1 subunit, which facilitates binding to the ACE2 receptor (angiotensin-converting enzyme 2) on the host cell. Interactions between the receptor-binding domain (RBD) in subunit S1 and the ACE2 receptor lead to large-scale structural rearrangements of the S protein, which is essential for virus cell entry (Figure 1b) [9,10,11]. The S protein further exhibits a unique peculiarity, namely, a furin-like cleavage site at the amino acidic positions 680–683 [12], which is considered to be responsible for easier priming of the S protein. This SARS-CoV-2-specific structural mechanism is purported to accelerate the spread of SARS-CoV-2 compared to other coronaviruses [12]. The N protein is the most abundant structural protein in SARS-CoV-2 and is crucial for viral genome replication and modulation of cell signaling pathways. The N protein is highly immunogenic [13], but antibodies to the viral N protein decline faster than those to the receptor-binding domain or the entire Spike protein and therefore may substantially underestimate the proportion of SARS-CoV-2-exposed individuals [14]. Both proteins, N and S, of SARS-CoV-2 are widely implemented for specific serological detection in various immunoassay methods [15,16,17].

All viruses ensure their survival and escape from the immune system through mutations in their genome. Thousands of SARS-CoV-2 mutations are circulating globally. Still, due to the proof-reading activity of the SARS-CoV-2′s RNA polymerase, the frequency of mutations is low in comparison to, for example, the human immunodeficiency virus (HIV), which lacks proof-reading activity [19]. Most of the newly formed variants of SARS-CoV-2 are inconsequential, but some may result in more infectious or threatening virus variants, the so-called variants of concern (VOCs) [20]. This can especially refer to variants containing mutations or deletions in the structural proteins of SARS-CoV-2 that may facilitate viral escape upon vaccination or reduce sensitivity towards serological tests [21,22]. Since 2019, an accumulation of mutations has occurred within SARS-CoV-2, leading to five WHO-classified specific new viral strains so far [20]. Considering mutations within the Spike-RBD protein, seven amino acid exchanges occurred in the early formed *Alpha* B.1.1.7 variant. Up to 30 amino acid exchanges and deletions have been reported within the Spike structure of the recent *Omicron* B.1.1.529 variant of SARS-CoV-2 (see Figure 2 for a detailed overview). These modifications within the Spike structure can, for example, modify the cleavage site of the furin protease of the host cell, making *Omicron* less dependent on target cells expressing the ACE2 receptor. This in turn results in less restriction to lung tissue, as well as a much higher reproduction factor and increased spreading. In contrast, *Omicron* is causing less severe infections and a lower rate of mortality [23]. 

As a variety of mutations and deletions are located within the Spike structure, especially within the RBD, of SARS-CoV-2, the immunogenicity of the Spike/RBD antigen is modulated. Various vaccines and diagnostic assays are based on the antigen structure of the wild type (wt) Spike or RBD of SARS-CoV-2 [24,25,26]. Concomitantly, the efficiency of vaccines and serological tests may be affected by these VOCs [27]. To counteract the lower efficiency of current vaccines, manufacturers such as BioNTech and Moderna have introduced a new generation of vaccines, where SARS-CoV-2′s wt S-RBD is supplemented with that of the B.4 and B.5 variants [28]. So far, there are several reports focusing on vaccines and VOC [29,30] and also some recent reports evaluating the impact of VOCs on diagnostic testing [31,32,33]. Until now there has been no systematic comparison of wt and variants of SARS-CoV-2 in terms of serological tests, especially on the PETIA-based assays.

Laboratory medicine has a critical role in the management of the pandemic. It is the key bottleneck for successful prevention, diagnosis, and treatment [34,35]. For initial diagnosis and monitoring of an acute SARS-CoV-2 infection, polymerase chain reaction (PCR) and viral antigen tests are used [36,37,38,39,40]. 

Despite their utility as direct diagnostic tools, these tests are not suitable for non-active infections and the determination of the disease’s prevalence (in a population) [41,42]. To identify patients who have overcome infection or to monitor the exposure of risk groups, the qualitative or quantitative measurement of SARS-CoV-2-specific antibodies in human blood is commonly used. Serological tests are also used to assess the patient-specific success of vaccination and the persistence of the vaccine-specific immune response [43]. Furthermore, detecting the neutralization potential of a patient’s antibody response and its linked individual protection in terms of variants is another important aspect. Antibody binding and ACE2 binding inhibition are significantly reduced for the *Omicron* variant compared to all other VOCs. Evaluation of serological anti-SARS-CoV-2 chemiluminescent immunoassays, correlated with live virus neutralization tests, for the detection of anti-RBD antibodies is a relevant alternative in COVID-19 large-scale neutralizing activity monitoring [44,45].

To date, various enzyme-linked immunosorbent assays (ELISA) or chemiluminescent immunoassays (CLIA) have been developed for the serological quantification of SARS-CoV-2. However, they have the major drawbacks of poor turnaround time and low throughput, reaching at most a couple of hundred tests per hour [46], and considerably high costs per test. These restrictions can have a strong limiting impact on their use, especially as the number of requested tests has dramatically increased due to vaccination campaigns. Particle-enhanced turbidimetric immunoassays (PETIAs) can solve these issues due to their applicability to a broad variety of routine clinical chemistry analyzers, easy handling, and reasonable cost structures. For this reason, PETIA-based serological assays are also successfully used in the quantification of a SARS-CoV-2-specific immunoresponse [44,47].

Realization of diagnostic tests in principle is a complex and long-lasting process with the need for balanced activities in design, development, and realization in production/scale-up, as well as validation and registration. Especially in the circumstances of a pandemic and the emergence of problematic variants, time-to-results are of absolute importance. 

Here, we present the design and performance evaluation of a Nucleocapsid- and RBD-based PETIA assay to quantify a SARS-CoV-2 immunoresponse. 

Furthermore, we evaluated the specificity and sensitivity of both PETIA immunoassays in parallel to investigate the theoretical bioinformatical assumption. A higher degree of similarity in the amino acid sequence of a protein can lead to a comparable immunoresponse and to subsequent cross-reactivity. This, in turn, may lead to the reduced specificity of the respective test based on this protein, as in the case of the SARS-CoV-2′s N protein. Selected antigens of the PETIA comprise the most sensitive and specific target structures that also have the lowest identity to other coronaviruses (RBD antigen displays 76% and Nucleocapsid protein 90.5% identity vs. the homologous protein of SARS-CoV). The suitability of the RBD-derived PETIA for its use in clinical routine was also investigated.

To ensure up-to-date serological assays, producers of immunoassays are required to constantly validate and improve their products with respect to new and potentially highly mutated variants of SARS-CoV-2 [48]. For this in this work, we propose a reproducible and easy-to-perform PETIA-quenching inhibition test for the evaluation of variant-derived SARS-CoV-2 antibodies. The PETIA-based inhibition assay and the presented rational bioinformatical design will also allow the rapid adaptation of immunoassays, which is essential to cover the ongoing diagnostic need of evolving VOCs in the progression of the pandemic.

## 2. Materials and Methods

### 2.1. Bioinformatical Characterization and Comparison of the Coronavirus Proteins and Structural Predictions

All protein sequences of SARS-CoV-2 and other related coronaviruses were derived from the UniProt (https://www.uniprot.org/, accessed on 12 May 2020) database (for the PubMed entry numbers, see Appendix A) and aligned using the software Multialin version 5.4.1 (Copyright I.N.R.A., France) [49]. The bioinformatical comparison of the different human infecting coronavirus Nucleocapsid (N), Spike (S), and Receptor Binding Domain (RBD) proteins was performed by Expasy SIM (https://web.expasy.org/sim/, accessed on 23 September 2020). A special focus was set to identify the lowest similarity among related viral proteins and to guarantee optimal specificity and sensitivity of antigen proteins or protein fragments.

The similarity was evaluated in terms of identity (%) and score, based on the following parameters:-Symbol comparison table: blosum62-Gap weight: 12-Gap length weight: 4-Consensus levels: high = 90%; low = 50%

Structural predictions of the Spike protein and the Receptor binding domain of SARS-CoV-2 were performed with the program “AlphaFold 2.2.4” (DeepMind, Google LLC, Mountain View, CA, USA), based on the UniProt P0DTC2 annotation of the SARS-CoV-2 protein sequence [50]. The representative structure reported in this work showed the highest confidence prediction among the five calculated models. Visualization was conducted by the software PyMOL 2.5 (https://pymol.org/, accessed on 2 October 2022).

### 2.2. PETIA Production

All PETIAs were produced by covalently coupling the viral antigens S-RBD (P-307-100) or N (P-301-100) of SARS-CoV-2 (Icosagen, Tartu, Estonia) to polystyrene beads (latex, produced by Merck (Pithivier, France) and Bangs Laboratories (Fishers, IN, USA)). The Lx-beads were washed three times in MES 100 mmol/L, pH 6.1. After the last wash, precipitated Lx-beads (26,000 rpm/min) were finally resuspended to a concentration of 2% and activated with 2 mmol/L EDC (1-Ethyl-3-(3-dimethylaminopropyl) carbodiimide Hydrochloride (Merck, Darmstadt, Germany) and NHS (N-hydroxysuccinimide, Merck, Darmstadt, Germany), respectively, while gently mixing. The activation reaction was allowed to proceed for 15 min at room temperature. After two washing steps in MES 100 mmol/L, pH 6.1, the desired amount of protein (80 µg of S-RBD or N protein from SARS-CoV-2, respectively) was added and left to react for 4 h at room temperature. The resulting latices were quenched and washed in glycine buffer 20 mmol/L, pH 8.0 (Merck, Darmstadt, Germany) and finally resuspended in glycine buffer, 20 mmol/L, pH 9.0 and BSA (bovine serum albumin) 1 g/L for further storage at 4 °C (R2) [51,52].

PETIA-based assays were formulated as two component reagents containing the functionalized, antibody-coated nanoparticles in reagent 2 (R2) and stabilizers and enhancing components in reagent 1 (R1). The reaction buffer (R1) was composed of TRIS 100 mmol/L, pH 6.5 (Merck, Darmstadt, Germany), containing 0.1% Tween-20 (Merck, Darmstadt, Germany) and 1% of the reaction enhancer polyethylene glycol 6000 (PEG6000, Merck, Darmstadt, Germany).

For evaluation, three specimens containing recombinant anti-SARS-CoV-2 antibodies were used (concentrations 150, 75, and 30 AU/mL, DiaServe Laboratories GmbH, Iffeldorf, Germany).

A BioMajesty™ JCA-BM6010/C fully automated clinical chemistry analyzer (JEOL Ltd., Tokyo, Japan) was utilized for testing reagent performance according to the following application:-Reaction buffer (R1) = 90 µL-Sample = 10 µL-Latex bead reagent (R2) = 30 µL-Wavelength = 658 nm-Reading time = ca. 600–350 s-Result calculation = Δ [Abs_600sec_ − Abs_350sec_]

### 2.3. Performance Comparison of S-RBD and N-Based PETIA

To evaluate the performances of the two PETIAs to a CLIA reference test, a direct comparison of 43 SARS-CoV-2 positive and 52 negative serum samples was conducted using a BioMajesty™ JCA-BM6010/C analyzer. Serum samples were collected from participants with a positive result of SARS-CoV-2 RT-PCR in a nasopharyngeal swab, at least 10 days before serum collection. For reference and comparison, the serum samples were additionally quantified by the Elecsys^®^ Anti-SARS-CoV-2 ECLIA test on a *cobas e 411* analyzer platform (Roche Diagnostics GmbH, Mannheim, Germany). The data were graphically represented using the tool “Data Comparison Graphs” of MedCalc Statistical Software version 18.10.2 (MedCalc Software bvba, Ostend, Belgium; http://www.medcalc.org; 2018, accessed on 29 November 2021).

### 2.4. Quantification of Variant Cross-Reactivity 

Purified (6 His-Tag) recombinant S-RBD antigens of SARS-CoV-2 (wild type, *Alpha* (B.1.1.7), *Beta* (B.1.351), *Gamma* (P.1), *Delta* (B.1.617.2), *Kappa* (B.1.617.1), and *Omicron* (B.1.1.529)), comprising amino acids (319-541) of S-RBD, were derived from Icosagen, Tartu, Estonia. Quantification and purity of the respective antigens were verified by SDS–PAGE and Coomassie staining (>95%). An equal amount of VOC S-RBD antigens, normalized to wt S-RBD antigen, was spiked into samples containing antibodies against SARS-CoV-2. To assess the kinetics of binding of the respective S-RBD antigens toward the SARS-CoV-2 specific antibodies of the sample, different concentrations of the antigens were used (100, 80, 60, 40, 30, 20, 10, 5, 2.5, 0 ng, see Appendix A). For the *Omicron* variant, higher amounts of the S-RBD antigen (500 and 1000 ng) were spiked to further confirm its lower inhibitory potential (see Appendix A). After 15 min of incubation and quenching, the residual binding reactivity of the preincubated samples was assessed by subsequent quantification of the remaining anti-SARS-CoV-2 antibodies with the S-RBD-based PETIA reagent using a BioMajesty™ JCA-BM6010/C fully automated clinical chemistry analyzer (JEOL Ltd., Tokyo, Japan) (*n* = 3). The obtained anti-SARS-CoV-2 concentrations were visualized, related to the not-quenched reference (0 ng antigen, 100% recovery/blank). The data were additionally graphically represented using the tools “Multiple Variables Graph” and “Multiple Lines Graph” of MedCalc Statistical Software version 18.10.2 (MedCalc Software bvba, Ostend, Belgium; http://www.medcalc.org; 2018, accessed on 29 November 2021).

## 3. Results

### 3.1. Bioinformatical Comparison of Coronavirus Proteins

The specific detection and quantification of SARS-CoV-2, unaffected by any unspecific cross-reactivity with other coronaviruses, is a prerequisite for serological SARS-CoV-2 test systems. To determine the degree of similarity between the protein sequences of the Spike structure, the S-RBD and the N protein of the related coronaviruses SARS-CoV-1, MERS-CoV, HKU1-CoV, OC43-CoV, and the *Alpha* coronaviruses 229E-CoV and NL63-CoV were aligned and compared (Table 1). The lowest similarity among viral proteins was observed for the S-RBD protein, enabling a clear immunological differentiation to other related coronaviruses when the S-RBD protein fragment is used for manufacturing a SARS-CoV-2 specific serological assay. All related coronaviruses showed less than 27.3% sequence similarity, compared to 37.3% similarity of the total S protein or 50.9% of the N protein.

### 3.2. Production and Performance Comparison of S-RBD vs. N-Based PETIA

The S-RBD of SARS-CoV-2 was bioinformatically identified as the most specific immunogenic structure among the coronaviral proteins (Table 1). To verify the bioinformatical prediction, two PETIA assays, one based on the S-RBD and another based on the N-protein of SARS-CoV-2, were produced in parallel. In total, 43 SARS-CoV-2 positive and 52 negative patient serum samples were directly compared by these two PETIA reagents on a BioMajesty™ JCA-BM6010/C fully automated clinical chemistry analyzer (Figure 3a,b). This analysis confirmed that the RBD protein does ensure better specificity (98%) and good sensitivity (93%) over the N protein (specificity 21%, sensitivity 100%) (see Appendix A). Classification of samples (COVID-19 negative/positive) was done by preceding nasopharyngeal RT-PCR tests of respective donors and also by measuring corresponding serum samples with the N-based Roche Elecsys^®^ Anti-SARS-CoV-2 CLIA reference test. The detailed serological results of the N-based Roche Elecsys^®^ Anti-SARS-CoV-2 CLIA and the two PETIAs (S-RBD and N-based) are reported in the Appendix A. 

### 3.3. Evaluation of Variant Cross-Reactivity

Alongside the overall performance (sensitivity and specificity) of a wt RBD-based SARS-CoV-2 immunoassay, the ability of the assay to also quantitatively recognize antibodies derived from mutated forms of SARS-CoV-2 (VOC) is essential. To this end, a PETIA-based inhibition test was used to assess the kinetics of binding and inhibition of six SARS-CoV-2 S-RBD variants (*Alpha*, *Beta*, *Delta*, *Gamma*, *Kappa*, and *Omicron*) in relation to the wt antigen S-RBD protein. Recombinant S-RBD antigens of SARS-CoV-2 (wild type, *Alpha* (B.1.1.7), *Beta* (B.1.351), *Gamma* (P.1), *Delta* (B.1.617.2), *Kappa* (B.1.617.1), and *Omicron* (B.1.1.529)) were thoroughly quantified and characterized to normalize the antigen protein content (Figure 4a). The titration of VOC antigens was used to assess the reactivity towards a defined sample, containing antibodies raised on a wt SARS-CoV-2 antigen-induced immunoresponse (representative kinetics are shown in Figure 4b). As evidenced by the inhibition dynamics, wt S-RBD antigen, as well as *Alpha*, *Beta*, *Gamma*, *Delta*, and *Kappa*, revealed an overall comparable reactivity. Showing a clear difference, the *Omicron* variant (B.1.1.529) had significantly lower reactivity (approx. 50%) compared to wt and all other VOCs (Figure 5). The deviating inhibitory potential of the *Omicron* variant was confirmed additionally by using the S-RBD antigen at 5–10 fold higher quenching concentrations (Appendix A).

## 4. Discussion

The coronavirus pandemic, which is still ongoing, can be considered the biggest challenge faced by humanity for many decades. In this scenario, laboratory medicine plays a central role [34,35]. In particular, serological tests, measuring anti-SARS-CoV-2 antibodies upon infection or after vaccination cycles, need to have optimal sensitivity and specificity, be robust and easy to use, and have a suitable cost structure.

The first goal of this work was to determine bioinformatically which SARS-CoV-2 protein is the most suitable for the development of an innovative PETIA-based immunoassay. The intrinsic sequence similarities of the corona-viral proteins can have a strong effect on the immunoassay cross-reactivities, resulting in a low specificity. This aspect has to be carefully considered in the design phase, since the correct choice of targeted protein can avoid or reduce the problem. Cross-reactivity of a SARS-CoV-2 immunoassay can arise from previous infections, especially for the HCoV-OC43, HCoV-HKU1, HCoV-NL63, and HCoV-229E viruses, responsible for typical colds. 

We compared the main structural proteins (N protein, S protein, and S-RBD) of the most common human-infecting coronaviruses and SARS-CoV-2. Our comprehensive in silico sequence comparison indicated that the S-RBD should be the first choice to guarantee the best assay specificity (Figure 3, Table 1). Indeed, the highest identity was observed between SARS-CoV-2 and SARS-CoV-1, whose similarity amounted to 90.5% for the N protein, 76% for the S, and 73.1% for the S-RBD. Furthermore, in light of SARS-CoV-1 probably being extinct [53,54], the relatively high similarity of both viruses raises no particular concern for the development of a SARS-CoV-2-specific serological assay. The protein alignments between SARS-CoV-2 and the other coronaviruses showed even lower degrees of similarity (Table 1). Additionally, in these cases, the S-RBD presented the lowest identity towards SARS-CoV-2 (vs. MERS-CoV = 24.7%; vs. HCoV-OC43 = 27.3%; vs. HCoV-HKU1 = 24.5%). These data are in concordance with previous literature [8,31,55] and indicate that from a bioinformatical point of view, the S protein and even its receptor binding domain (RBD) are the best choice to obtain the lowest cross-reactivity and the highest specificity when utilized as an antigen. The immunoassay presented here has been rationally designed, as the selection of the SARS-CoV-2 antigen to be used was based on a comprehensive a priori bioinformatical analysis.

A major challenge in assay development is the homogenous principle of PETIAs, which can be more prone to unspecific reactions than heterogeneous ones, such as ELISA or CLIA technologies. Here, we present for the first time the parallel development of an N vs. an S-RBD antigen-based SARS-CoV-2 assay, based on the same technology (PETIA). A direct comparison of the diagnostic performances of both assays was performed with 95 serum samples, previously analyzed with PCR molecular diagnostics and a CLIA reference assay. The obtained data clearly confirmed the theoretical assumptions: the S-RBD-based PETIA has a specificity of 98% vs. 21% for the N-based one. The observed lower sensitivity of the RBD-based immunoassay can be explained by the use of the N-based tests from Roche as a reference assay. This observation is supported by the sensitivity of the N-based PETIA (100%). Likely, some donor samples also contained anti-N antibodies but not (yet) specific anti-RBD ones. Indeed, Smits et al. reported that the whole N protein sequence could induce the humoral response, while only a specific peptide of the S protein seemed to be significant in this [56]. The evident overlap between the positive and negative sample cohorts can be further explained by the sequence similarity of corona-viral N proteins. This similarity may lead to a possible cross-reactivity of the human antibodies of individual samples when patients have been exposed to a coronavirus-based previous infection (see the sequence comparison in Table 1). However, the magnitude of overlap seems to suggest a cooperative effect of unspecific reactions, proving that an N protein-based PETIA is not suitable for diagnostic purposes. On the other hand, the S-RBD protein-based PETIA reagent showed remarkable performance, and its specificity (98%) did not seem to be influenced by cross-reactivities or the homogenous assay technique.

However, it is important to emphasize that the serological tests that are exclusively based on the S-RBD antigen cannot distinguish among antibodies originating from a SARS-CoV-2 infection or vaccinations using the S-RBD antigen. To assess previous infections in vaccinated patients, the additional use of the N-based serological test is suggested. The highly abundant and immunogenic N protein causes an intensive immunoresponse upon infection, which in turn indicates that this antigen could lead to superior sensitivity, as was also previously reported by Burbelo et al. [13]. The superior sensitivity of the N antigen was also confirmed for the N-based PETIA technology correlating 43 RT-PCR positive COVID-19 samples to the Roche CLIA Elecsys^®^ Anti-SARS-CoV-2 reference assay (see Figure 3b). Indeed, both these N-based tests showed a sensitivity of 100%.

The obtained performance characteristics showed that specific and sensitive SARS-CoV-2 immunoassays based on PETIA technology can be designed and produced in a very efficient and rapid manner. Validation on 95 pre-defined reference samples (Figure 3) further supports the suitability of the RBD-based PETIA for its use in the clinical routine laboratory to quantify anti-SARS-CoV-2 antibodies in human samples upon infection or vaccination.

The homogenous PETIA technology, lacking the removal of the unbound sample and additional washing steps, could potentially lead to a lower specificity in comparison to the heterogenous CLIA or ELISA technologies. Our data show that the specificity of the S-RBD-based PETIA is 98%, and it is well comparable to CLIA-based technology (Figure 3a). On the other hand, PETIA sensitivity is mainly affected by the antigen used for the assay setup, more than by the technology itself (CLIA or PETIA). Furthermore, the majority of available serological CLIA- and ELISA-based assays can quantify only one isotype of antibody (usually IgG, due to the used secondary detection antibody), while the PETIA can react with all isotypes, including the early secreted IgM and IgA. This aspect leads to an earlier detection of the immunoresponse by the PETIA technology and to a better sensitivity, especially at the onset of an infection [47].

This finding was also supported by the recent work of Spaeth et al. and Brehm et al., indicating the applicability of PETIA technology to common clinical chemistry laboratory devices and the superior time to obtain results compared to CLIA and ELISA-based techniques (approx. 10 min vs. 30 min vs. 150 min) [44,47].

To date, thousands of single mutations or modifications (insertions/deletions) within the coding genes of SARS-CoV-2 have been described [57]. Most of the newly formed variants of SARS-CoV-2 are harmless, or lack immunogenic importance, and do not affect vaccines or serological assays. In contrast, variants that contain mutations or deletions in the structural proteins of SARS-CoV-2, especially in the very prominent Spike structure and/or its dominant S-RBD, facilitate viral escape upon vaccination [57] or modulate sensitivity towards serological tests. All vaccines, as well as nearly all commercial diagnostic tests, are based on the antigen structure of the wt Spike protein or the S-RBD of SARS-CoV-2. Consequently, the wt Spike/RBD antigen used in immunoassays may not quantitatively recognize the antibody population in patients when it is raised upon infection by variants of concern of SARS-CoV-2. To guarantee quantitative results of a SARS-CoV-2 specific serological assay, a clear understanding of the impact of VOC-derived antibodies on immunoassays utilizing wild type-based antigens is necessary. 

In this work, the cross-reactivity of the RBD-based immunoassay was assessed by a quenching inhibition test. The kinetics of binding and blocking of the wt SARS-CoV-2′s S-RBD and the six derived variants of concern, *Alpha*, *Beta*, *Delta*, *Gamma*, *Kappa*, and *Omicron*, were analyzed. The PETIA utilizes a recombinantly expressed wt fragment of the S-RBD antigen bound to the functionalized nanoparticle surface. For this, to directly compare the reactivity of VOC antigens within the same PETIA format, constructs of recombinant fragments of respective VOCs were used in an inhibitory quenching assay. Integrity, purity, and concentration of the respective fragments were evaluated and normalized to the wt S-RBD through SDS–PAGE analysis (Figure 4a). In all variants, the S-RBD-based PETIA was inhibited (Figure 4b). This demonstrated that the wt S-RBD-based PETIA overall can detect cross-reactive variant-derived antibodies to a significant extent. By the obtained inhibitory activity, it is evident that the variants *Alpha*, *Beta*, *Delta*, *Gamma*, and *Kappa* share the same inhibition kinetics, and thus they can be quantified to the same extent as wt-derived antibodies. In clear contrast is the *Omicron* variant. In the same setting, its inhibitory activity was less pronounced, indicating that *Omicron*-derived antibodies likely cannot be quantified to the same extent as the wt antigen or the other tested variants (Figure 4b and Figure 5). Although the *Omicron* variant seems to keep an intrinsic binding and inhibiting effect to some extent, the S-RBD-based PETIA can only be used for semi-quantitative diagnostics of *Omicron*-derived antibodies. The Spike structure of SARS-CoV-2 is highly dynamic and prominently exposed to the host target cells (Figure 6a). For this, the S-RBD structure is highly immunogenic and in turn also accessible to derived neutralizing antibodies [56]. The RBD fragment, used as the PETIA assay antigen, comprises 222 amino acids spanning 319–541 (Figure 6b).

In the *Omicron* variant B.1.1.529, three mutations are located within the peripheral, flexible loop region of the receptor binding domain, putatively altering the immunogenicity of *Omicron* SARS-CoV-2 (Figure 6b, highlighted in yellow and indicated by asterisks). Direct mutations within the SARS-CoV-2’s S-RBD, as well as further mutations in the Spike protein, lead to an altered overall appearance of the *Omicron* Spike in patients and a mixed population of derived antibodies. This mix represents antibodies that share epitopes with the wt Spike but also ones that are unique to *Omicron*. These structural aspects well explain the reduced reactivity of approx. 50% observed in the inhibitory quenching assay with the mutated *Omicron* form of SARS-CoV-2. 

The rational bioinformatical approach presented here can enable the fast and efficient design of serological assays. Due to the growing availability of dedicated sequence information from new VOCs, the bioinformatical comparison in combination with structural predictions will make it possible to identify regions of the SARS-CoV-2 spike protein that are less susceptible to mutations or in which the mutations do not cause a significant change in the immunogenicity of the protein. This bioinformatical information enables the better and faster identification of conserved, VOC-invariant Spike structures for broad-based antibody recognition, independent of the respective VOC antigen. On the other hand, specific immunoassays against individual mutations that do not cross-react with the wild type or other mutants can also be developed very quickly using bioinformatical approaches. Comparative bioinformatical information therefore will facilitate the rapid adjustment of the PETIA to putative future different VOC antigens, showing a reduced or no cross-reactivity with the wild type antigen (for example in the case of the *Omicron* variant, Figure 4 and Figure 5). This aspect is of utmost importance, as immunoassay producers are expected to constantly prove the suitability of their tests for new variants [48]. The assessment of the binding kinetics of different VOC antigens that we proposed here by the inhibition quenching test supplied a quantitative and easy-to-perform cross-reactivity methodology. It is noteworthy that this method could be reliably reproduced also to test the suitability of the immunoassays for future VOCs. Although the PETIA technology enables the rapid adaptation of the assay setup and thus a VOC-specific quantification, the initial quantitative one-step differentiation of wild type or other VOCs is not possible. For this purpose, the other known VOCs would need to be assessed simultaneously, for example, in a multiplexing procedure, and be referenced to the wt S-RBD antigen. The increasing information on a multitude of VOCs (see Figure 2 and Figure 6) and the bioinformatical approach presented here will enable the in-time identification of these kinds of invariant Spike antigen structures of SARS-CoV-2.

In summary, in this work, a direct comparison of the SARS-CoV-2 N and S-RBD proteins as antigens for the development of serological reagents was performed. Two PETIAs were designed and produced in parallel, utilizing an innovative approach featuring bioinformatics. This work proves the theoretical bioinformatical assumptions of the superior diagnostic performance of an S-RBD-based assay, whereas the N-based PETIA did not show a sufficient performance for diagnostic purposes. The S-RBD-based assay showed specificity and sensitivity suitable for its clinical routine use, in comparison to a CLIA test already present on the market. 

Finally, this work showed that PETIA technology is suitable for the serological testing of SARS-CoV-2 variants of concern. By the use of a quenching inhibition test, it has been proven that antibodies derived from five SARS-CoV-2 S-RBD-variants (*Alpha*, *Beta*, *Delta*, *Gamma*, and *Kappa*) can be well quantified using the S-RBD-based PETIA. On the contrary, *Omicron* variant-derived antibodies are significantly less recognized by wt antigen S-RBD-based PETIA. However, the bioinformatical approach proposed here enables the rapid and efficient analysis of the Spike proteins of different VOCs and a fast and reliable adjustment of the respective immunoassays.

## 5. Patents

T.M., L.W., C.K. and M.G. are named as inventors on a patent application (International patent application WO 2022/043147 A1), claiming the manufacturing and use of the described RBD-based PETIA for serological quantification of SARS-CoV-2 antibodies. The PETIA SARS-CoV-2 UTAB FS assay used in this study was kindly provided by DiaSys Diagnostic Systems GmbH, Holzheim, Germany.

## Figures and Tables

**Figure 1 biomedicines-11-00160-f001:**
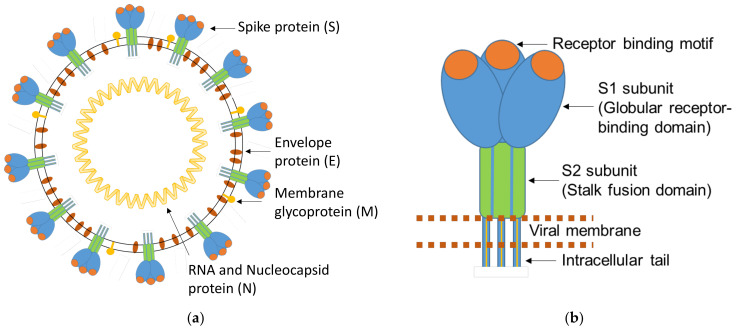
(**a**) Schematic representation of a SARS-CoV-2 virion; (**b**) enlarged scheme of the trimeric structure of the SARS-CoV-2 S protein. Adapted from Mittal et al. [18].

**Figure 2 biomedicines-11-00160-f002:**
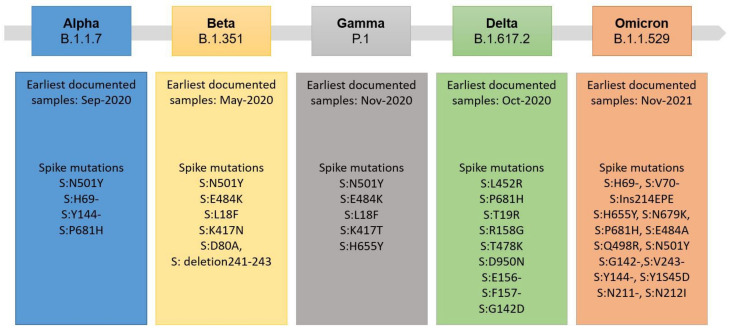
Overview of SARS-CoV-2 variants and notable Spike mutations, which were declared variants of concern (VOC) by the WHO [20,21].

**Figure 3 biomedicines-11-00160-f003:**
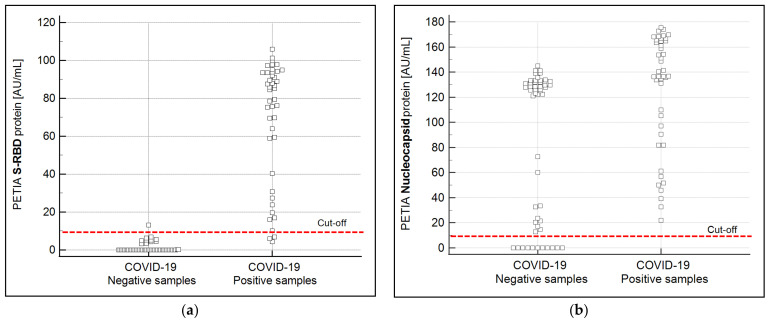
Graphical representation of the sample recovery of the S-RBD (**a**) and the N (**b**) protein-based PETIA. The overlapping of the negative and the positive samples for the N-based PETIA is consistent with the theoretical assumptions of the bioinformatical finding that N protein-based immunoassays reveal more unspecific cross-reactivity. The dashed red line indicates the cut-off of the used PETIA tests.

**Figure 4 biomedicines-11-00160-f004:**
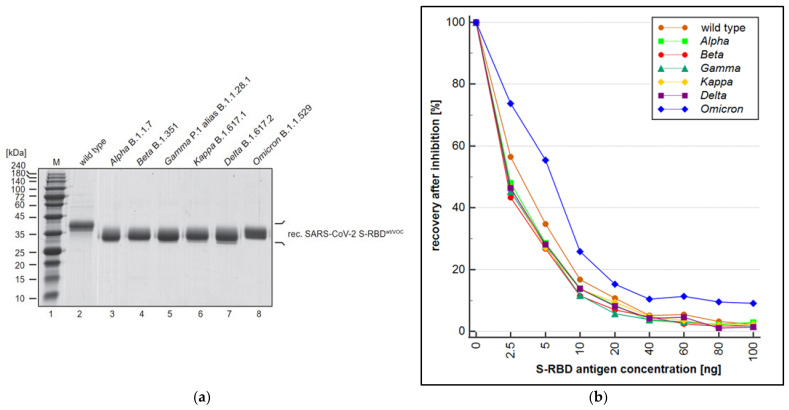
(**a**) Quantity and purity of the recombinant S-RBD fragment of SARS-CoV-2 wild type (lane 2), *Alpha* B.1.1.7 (lane 3), *Beta* B.1.351 (lane 4), *Gamma* P.1 alias B.1.1.28.1 (lane 5), *Kappa* B.1.617.1 (lane 6), *Delta* B.1.617.2 (lane 7), and *Omicron* B.1.1.529 (lane 8) were evaluated by SDS–PAGE and Coomassie staining; (**b**) S-RBD antigen concentration in the reaction in ng is shown on the *x*-axis (mean; *n* = 3), plotted against the recovery of each antigen in % related to a non-quenched sample. In comparison to the wild type, all evaluated variants except the *Omicron* variant showed comparable or lower quantification.

**Figure 5 biomedicines-11-00160-f005:**
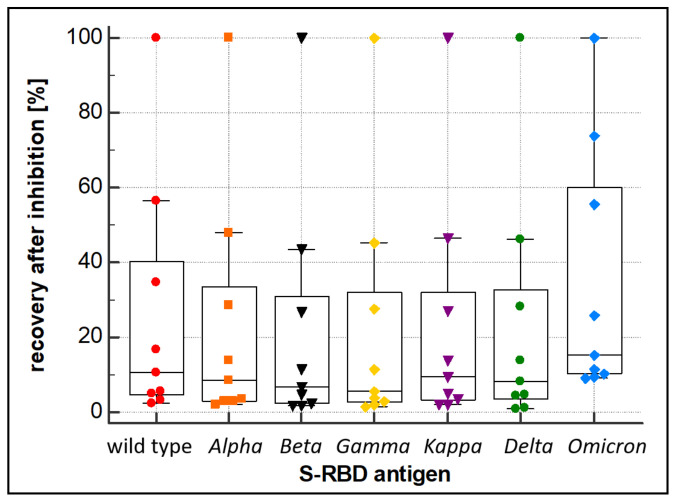
Inhibition behavior of the wild type, *Alpha*, *Beta*, *Gamma*, *Kappa*, *Delta*, and *Omicron* S-RBD variants. Inhibition was evaluated by quantification of an anti-SARS-CoV-2 antibody-containing sample after quenching with the recombinant wild type, *Alpha*, *Beta*, *Gamma*, *Kappa*, *Delta*, and *Omicron* S-RBD variant proteins, respectively.

**Figure 6 biomedicines-11-00160-f006:**
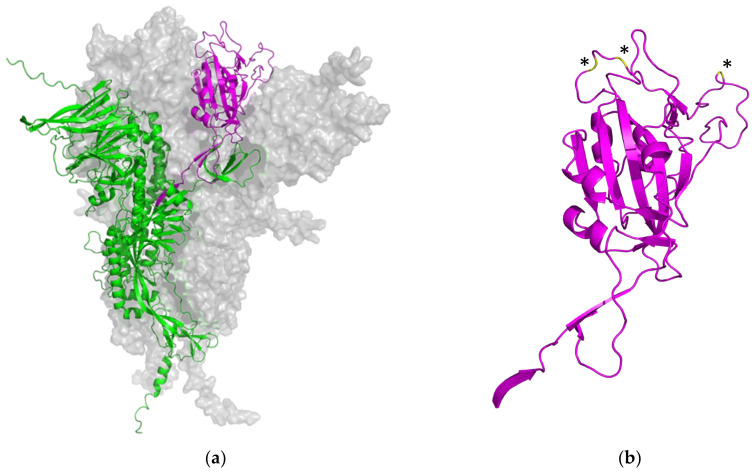
(**a**) Overall representation of the homotrimeric Spike structure of SARS-CoV-2 (two homomers are shown as light gray surfaces). One monomeric structure is shown in more detail as a ribbon representation. The ACE2 receptor binding domain is shown in magenta, and the N-terminal domain, the general structure, the central helix, and the connector domain are shown in green. (**b**) Ribbon representation of wild type S-RBD domain 314–541 used for the S-RBD PETIA (magenta). Omicron B.1.1.529 mutations (E484A, Q498R, N501Y) within the S-RBD are highlighted in yellow and indicated by asterisks. Predictions by AlphaFold 2.2.4, based on the UniProt P0DTC2 SARS-CoV-2 protein sequence.

**Table 1 biomedicines-11-00160-t001:** Comparison of the Spike-RBD (2nd column), Spike (3rd column), and Nucleocapsid proteins (4th column) of SARS-CoV-2 vs. the other six human infecting coronaviruses. The evaluation is reported in terms of sequence similarity (%) and score (in parentheses). * The similarity between/these two proteins was too low and the comparable sequence was too short to report significant values.

Virus	Identity of Related Viral Proteins to SARS-CoV-2 (%) (Score)		
	S-RBD SARS-CoV-2	S Protein SARS-CoV-2	N Protein SARS-CoV-2
SARS-CoV-1 *	73.1% (896)	76.0% (5119)	90.5% (1993)
MERS-CoV	24.7% (86)	34.1% (1249)	50.9% (855)
HCoV-OC43	27.3% (130)	37.3% (1108)	42.0% (397)
HCoV-HKU1	24.5% (112)	35.8% (1040)	36.7% (424)
HCoV-NL63	- * (42)	33.3% (706)	32.1% (260)
HCoV-229E	- * (35)	34.4% (754)	26.4% (153)

## Data Availability

All data are available upon approval of the corresponding author.

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
