# Peer review of "Bioinformatical Design and Performance Evaluation of a Nucleocapsid- and an RBD-Based Particle Enhanced Turbidimetric Immunoassay (PETIA) to Quantify the Wild Type and Variants of Concern-Derived Immunoreactivity of SARS-CoV-2"

_biomedicines, 2023, doi:10.3390/biomedicines11010160_

Round 1
Reviewer 1 Report
I think the PETIA method designed for rapid antibody testing is an idea that will be used as a very useful method for antibody screening.
I can appreciate the excellence of the methodology, but I wonder if it is available in a way that reflects the antigenic properties of SARS-CoV-2, which is changing very quickly.
However, the specificity of the method is very different, with S-RBD being 98% while N protein being 21%. Therefore, in the case of S-RBD, it should be described in a way that only the antibody levels of S-RBD can be seen as a site where antibodies by vaccines and antibodies by infection cannot be distinguished.
In addition, the weak response to the omicron variant is that the response to the newly mutated variant may change in the future, and a description of how to respond to these problems is needed.
Author Response
Reply to Review biomedicines-2106354, Review 1
Dear Reviewer,
Thank you very much for your suggestions and valuable comments.
To improve our manuscript´s quality, we used the MDPI´s Language Editing Services. The English of the paper has been now officially proofed and spell-checked by a native speaker.
We have now addressed your raised concerns in the revised version of the manuscript, uploaded here for your review. All changes made are marked up accordingly in red color. Please find also our point-by-point comments, directly attached to your raised points.
We hope that the improvements and modifications listed here will now allow the manuscript to be positively reviewed by Biomedicines.
Comment 1: “I can appreciate the excellence of the methodology, but I wonder if it is available in a way that reflects the antigenic properties of SARS-CoV-2, which is changing very quickly.”
Answer 1: The rational bioinformatical approach presented here enables the fast and efficient design of serological assays. It also facilitates the rapid adjustment of the PETIA to putative future different VOC antigens, showing a reduced or no cross-reactivity with the wild type antigen (e.g. in the case of the Omicron variant). This aspect is of upmost importance as the immunoassay producers are expected to constantly prove the suitability of their tests for new variants. The assessment of the binding kinetics of different VOC antigens that we propose here by the inhibition quenching test, does supply a quantitative and easy-to-perform cross-reactivity methodology. It is noteworthy, that this method can be reliably reproduced also to test the suitability of the immunoassays for future VOCs. These new aspects and new references are now integrated to the discussion part of the revised manuscript (lines 964-982).
Comment 2: “However, the specificity of the method is very different, with S-RBD being 98% while N protein being 21%. Therefore, in the case of S-RBD, it should be described in a way that only the antibody levels of S-RBD can be seen as a site where antibodies by vaccines and antibodies by infection cannot be distinguished.”
Answer 2: This is a very valid point, raised by Reviewer 1. Serological tests that are exclusively based on the S-RBD antigen cannot distinguish among antibodies originated from an SARS-CoV-2 infection or rised upon vaccinations by this antigen. To assess previous infections in vaccinated patients, the additional use of the N-based serological test is required. The highly abundant and immunogenic N protein causes an intensive immunoresponse upon infection, which in turn indicates that this antigen could lead to a superior sensitivity, as also previously reported by Burbelo et al. The superior sensitivity of the N antigen was also confirmed for the N-based PETIA technology by correlation of the 43 RT-PCR positive COVID-19 samples to the Roche CLIA Elecsys® Anti-SARS-CoV-2 reference assay. This information as well as the additional references are now included in the revised version of the dicussion (lines 851-861). Moreover, more detailed data on the serological test sensitivity is now added in supplemental table 7.
Comment 3: “In addition, the weak response to the omicron variant is that the response to the newly mutated variant may change in the future, and a description of how to respond to these problems is needed”
Answer 3: The cross-reactivity of the immunoassays with possible new variants is of upmost importance and the producers are requested to check it for their own tests. The methodology we report here facilitates this analysis, via the use of a simple inhibition quenching test, and the easier adaptation of the PETIA technology to new variants, via the employment of the bioinformatical approach. We highligthed these aspects in the revised discussion (lines 964-973). Although the PETIA technology enables the rapid adaptation of the assay setup and thus a VOC-specific quantification, an initial quantitative one-step differentiation of wild type and other VOCs is not possible. For this purpose, the other known VOCs would need to be assessed simultaneously, e.g. in a multiplexing procedure, and to be referenced to the wt S-RBD antigen. Beside multiplexing technology, another possibility to assess a VOC-invariant immunoresponse is the identification of S-RBD specific antigen-motives or a combination of respective peptides that are not, or less sensitive to mutations. These new aspects are now intergrated into the discussion part of the revised manuscript (line 973-982).
Comment 4: Does the introduction provide sufficient background and inlcude all relevant references? – Can be improved
Answer 4: Additional literature and more detailled explanations on diagnostic needs an possible solutions concerning the VOCs were integrated into the revised version of the manuscript (lines 92 and 254-261).
Comment 5: Are all the cited references relevant to the research? – Can be improved
Answer 5: All literature sources have been once again cross-checked and also specific references where explaind in more detail in the mamnuript.
Best Regards,
Matthias Grimmler, on behalf of all authors
Reviewer 2 Report
The authors demonstrated the performance of the receptor-binding domain (RBD)-derived Particle Enhanced Turbidimetric Immunoassay (PETIA) for the serological test of SARS-CoV-2 infection. Overall, the descriptions are well described, which assists the readers’ understanding. This methodology is an innovative way to know the VoC status with serum samples from COVID-19 patients; however, the quantification for the omicron variant might be limited with S-RBD-based PETIA.
Were there no differences in the results of clinical samples between RT-PCR with the nasopharyngeal swab and the serum CLIA tests? A clear description of the association between the two tests would be appreciated in Lines 289 and 292. The benefits of PETIA include easy handling and reasonable cost structures, as the authors mentioned in the introduction. Is there a difference expected in the test performance of sensitivity and specificity between the PETIA and the other serological tests of ELISA/CLIA?
Author Response
Reply to Review biomedicines-2106354, Review 2
Dear Reviewer,
Thank you very much for your suggestions and valuable comments.
To improve our manuscript´s quality, we used the MDPI´s Language Editing Services. The English of the paper has been now officially proofed and spell-checked by a native speaker.
We have now addressed your raised concerns in the revised version of the manuscript, uploaded here for your review. All changes made are marked up accordingly in red color. Please find also our point-by-point comments, directly attached to your raised points.
We hope that the improvements and modifications listed here will allow the manuscript to be now positively reviewed by Biomedicines.
Comment 1: “This methodology is an innovative way to know the VoC status with serum samples from COVID-19 patients; however, the quantification for the omicron variant might be limited with S-RBD-based PETIA.”
Answer 1: This is a very important aspect. Our work underlines the importance of constantly checking the cross-reactivity of the immunoassays with possible new variants. This aspects is recommended also by the IFCC (International Federation of Clinical Chemistry) to all immunoassay producers. The inhibition test does offer a simple and at the same time robust analytical methodology to check the suitability of the immunoassays for new VOCs. Furthermore, the bioinformatical approach that we propose here, favours the adaptation of the PETIA technology to new variants, by simply identifying in silico the most suitable new antigen and using it to easily produce a modified PETIA as described in Materials & Methods. We now integrated these aspects to the revised version of the manuscript (lines 964-982). However, the direct on-step quantification of different muations (e.g. Omicron) is not possible by a PETIA assay. We now also illustrate on advantages but also limitations of the presented PETIA in this regard and discuss on other methods (e.g. multiplexing) to cover these topics in the revised version of the manuscript (lines 868-878 and 973-982).
Comment 2:
a) “Were there no differences in the results of clinical samples between RT-PCR with the nasopharyngeal swab and the serum CLIA tests? A clear description of the association between the two tests would be appreciated in Lines 289 and 292. The benefits of PETIA include easy handling and reasonable cost structures, as the authors mentioned in the introduction.”
b) “Is there a difference expected in the test performance of sensitivity and specificity between the PETIA and the other serological tests of ELISA/CLIA?”
Answer 2a: We included a more detailed description of classification of samples and quantification of serological methods to the methods part of the revised manuscript (lines 342-344) and in its discussion part (lines 857-861). Moreover, more detailed data on direct values and sensitivity of the serological test is now added in the new supplemental table 7.
Answer 2b: The homogenous PETIA-technology, lacking the removal of the unbound sample and additional washing steps, could potentially lead to a lower specificity in comparison to the heterogeneous technologies CLIA or ELISA. However, our data show that the specificity of the S-RBD-based PETIA is 98 % and it is well comparable to CLIA-based technology. On the other hand, the PETIA sensitivity is mainly affected by the antigen used for the assay set-up, more than by the technology itself (CLIA or PETIA). Furthermore, the majority of available serological CLIA- and ELISA-based assays can quantify only one isotype of antibody (usually IgG, due to the used secondary detection antibody), while the PETIA can react with all isotypes, including the early secreted IgM and IgA. This aspect leads to an earlier detection of the immunoresponse by the PETIA technology and to a better sensitivity, especially at the onset of an infection. These aspects are now integrated and discussed in the revised version of the manuscript (lines 868-878). New references were also added in the manuscript in this regard.
Comment 3: Does the introduction provide sufficient background and inlcude all relevant references? – Can be improved
Answer 3: The introduction was added with new literature and more detailled explanations about the problems concerning the VOCs and their solutions proposed in the paper (lines 92 and 254-261).
Best Regards,
Matthias Grimmler, on behalf of all authors
Round 2
Reviewer 1 Report
I think the direction of the study is well organized because the contents of the discussion have been modified a lot.
I think it explained the use of the reagent developed in this study and the limitations of the analysis of the results.
As an advantage of the bioinformational approach, if you can mention future changes to the appearance of mutants, it will be possible to set it up with a more convenient, rapid, and ready test method.
Author Response
Reply biomedicines-2106354, Review Round 2. Review 1
Dear Reviewer,
Thank you very much for your helpful comments and your positive overall impression of the revised manuscript, especially in regard of the intensively updated discussion.
The manuscript was checked by the MDPI´s Language Editing Services.
Now in the second round of revision a native speaking scientist reviewed the manuscript once again. Please find all changes of these two rounds of English editing activities highlighted in the revised version of the manuscript.
Comment 1: “As an advantage of the bioinformatical approach, if you can mention future changes to the appearance of mutants, it will be possible to set it up with a more convenient, rapid, and ready test method.”
Answer 1: As suggested by you, we further deepened the aspects connected to the appearance of new SARS-CoV-2 mutants and the use of our bioinformatical approach to enable the faster and easier in silico evaluation of the mutants and also the rapid testing of these new mutants by the inhibitory PETIA quenching test. These new aspects are integrated in the revised discussion, lines 490 to 502 and 532 to 535, and refer back directly to the presented results (especially Figure 4 and 5).
We hope that the improvements and modifications listed here will now allow the manuscript to be positively reviewed by Biomedicines.
Best regards,
Matthias Grimmler, on behalf of all authors